



# Kinetics of OH + SO₂ + M: Temperature-dependent rate coefficients in the fall-off regime and the influence of water vapour

Wenyu Sun,[1] Matias Berasategui,[1] Andrea Pozzer,[1] Jos lelieveld[1] and John N. Crowley[1]

Atmospheric Chemistry Department, Max-Planck-Institute for Chemistry, 55128 Mainz, Germany

*Correspondence to*: John N. Crowley (John.Crowley@mpic.de)

**Abstract.** The OH-initiated oxidation of SO₂ is the dominant, first step in the transformation of this atmospherically important trace-gas to particulate sulphate and accurate rate coefficients for the title reaction under all atmospheric conditions (pressures, temperatures and humidity) are required to assess its role in e.g. new particle formation. Prior to this study, no temperature dependent data were available in the fall-off regime for atmospherically relevant bath-gases. We thus address an important

omission in the kinetic database for this reaction and highlight significant discrepancies in recommended parameterizations. In this work, generation of OH via pulsed laser photolysis at 248 and 351 nm was coupled to its detection by laser induced fluorescence to obtain rate coefficients ($k_1$) for the title reaction at pressures of 14−742 Torr and temperatures of 220−333 K in N₂ bath gas. In-situ SO₂ concentrations, central to accurate kinetic measurements under pseudo-first-order conditions, were measured by optical absorption. Under the conditions of the present study, the termolecular reaction between OH and SO₂ is

in the fall-off regime and we parameterized the rate coefficients in N₂ in terms of low- ($k_{1,0}$) and high-pressure ($k_{1,\infty}$) limiting rate coefficients and a broadening factor ($F_C$) to obtain $k_{1,0}^{N2}$= 3.03×10⁻³¹ (T/300)⁻⁴·¹⁰ cm⁶ molecule⁻² s⁻¹, $k_{1,\infty}$ = 2.00 ×10⁻¹² cm³ molecule⁻¹ s⁻¹, and $F_C$ = 0.58. The effects of water vapour on the title reaction were explored through measurements in N₂-H₂O mixtures at 273, 298 and 333 K using the same experimental methods. The rate coefficients are significantly enhanced by the presence of water vapour with $k_{1,0}^{H2O}$ = 1.65×10⁻³⁰ cm⁶ molecule⁻² s⁻¹, which indicates that H₂O is a factor >5 more efficient in

quenching the HOSO₂* association complex than N₂. A model-based comparison of our rate coefficients and parameterization with previous literature measurements and recommendations of evaluation panels are presented and discussed. The use of the new parameterization instead of the IUPAC or NASA evaluations, particularly after including H₂O as a third-body quencher, leads to a significant (10-30%) reduction in the lifetime of SO₂ in some parts of the atmosphere and increases the H₂SO₄ / SO₂ ratio concomitantly.

## 1 Introduction

Sulphur enters the atmosphere predominantly in the form of gaseous sulphur dioxide (SO₂) which results from both natural and anthropogenic sources (Stevenson et al., 2003). The amount of SO₂ produced from human activities, particularly via fossil fuel combustion, is similar to that resulting from natural emissions (e.g. volcanic eruptions), and can be dominant on a regional scale (Brown, 1982; Brimblecombe, 2013; Lelieveld et al., 1997). SO₂ is a key intermediate during the complex chemical and





photochemical reactions that ultimately transform reduced sulphur compounds to sulphates. The oxidation of $SO_2$ in the atmosphere occurs in the gas phase as well as in droplets and aerosol particles (Liu et al., 2020; Cox, 1979; Beilke and Gravenhorst, 1978). The gas-phase oxidation of $SO_2$ is initiated mainly by the OH radical (R1), with a small contribution in forested regions by stabilised Criegee intermediates (Mauldin Iii et al., 2012; Huang and Chao, 2015). Based on a seasonal, global average boundary layer OH concentration of $1\times10^6$ molecule $cm^{-3}$ and the present recommendation for the rate

coefficient (Atkinson et al., 2004; Iupac, 2021), the lifetime of $SO_2$ with respect to reaction with OH (to form $HOSO_2$) is a few days.

$SO_2 + OH + M \quad\quad \rightarrow \quad\quad HOSO_2 + M$ (R1)

Once collisionally stabilized $HOSO_2$ reacts with $O_2$ to form $SO_3$ (R2), which is hydrolysed to sulphuric acid ($H_2SO_4$) (R3).

$HOSO_2 + O_2 \quad\quad \rightarrow \quad\quad HO_2 + SO_3$ (R2)

$SO_3 + (H_2O)_n \quad\quad \rightarrow \quad\quad H_2SO_4 \quad + (H_2O)_{n-1}$ (R3)

$H_2SO_4$ can initiate particle formation (e.g. via reactions with basic trace-gases such as $NH_3$) or condense on existing particles, thus contributing to aerosol formation, growth and cloud droplet nucleation (Kulmala et al., 1998; Vehkamäki et al., 2002; Sipilä et al., 2010; Saltzman et al., 1983). The aforementioned processes occur throughout the atmosphere, affecting ecosystems as well as the earth troposphere radiation budget and thus climate (Badr and Probert, 1994; Lelieveld and

Heintzenberg, 1992; Stevenson et al., 2003; Feichter et al., 1996). In addition, the oxidation of $SO_2$ to sulphate is a major sink of stratospheric OH and water (Bekki, 1995) and provides surface area for heterogeneous processes that e.g. contribute to stratospheric ozone depletion (Weisenstein et al., 1996; Heckendorn et al., 2009).

As a result of its central importance to atmospheric chemistry, the kinetics of the title reaction have been investigated in numerous experimental studies. The results of laboratory investigations of $k_1$, which serve as the basis for the IUPAC (Iupac,

2021) and NASA (Burkholder et al., 2020) evaluation panels, are summarized in **Table 1**. A few early measurements of $k_1$ at around 298 K and 1 atm $N_2$ (or air) (Izumi et al., 1984; Barnes et al., 1986; Davis et al., 1979; Castleman Jr and Tang, 1976; Cox and Sheppard, 1980) are not included in **Table 1**, as they do not contain information on the pressure- or temperature-dependence of the title reaction and display relatively large differences in $k_1$ at 298 K and 1 atm with values ranging from $5.99\times10^{-13}$ to $1.22\times10^{-12}$ $cm^3$ molecule$^{-1}$ s$^{-1}$. Surprisingly, for such an important reaction in atmospheric chemistry, no

temperature-dependent measurements of $k_1$ in the fall-off regime have been carried out in atmospherically relevant bath gases (e.g. air or $N_2$) but in He for which the collision efficiency is much lower than for $N_2$ and $O_2$, the dominant components of the atmosphere. In addition, the latest measurements of $k_1$, (Blitz et al., 2017a) using modern, laser-based photolysis methods, suggest that previous measurements were strongly biased (to larger values) by the photo-excitation of $SO_2$ (see later).

The title reaction is in the fall-off regime across the temperature/pressure ranges in the atmosphere. To parameterize the rate

coefficients for such reactions, the Troe-type formulation (Troe, 1983) is widely-used, which requires experimentally-determined high-pressure ($k_{1,\infty}$) and low-pressure ($k_{1,0}$) limiting rate coefficients as well as a broadening factor describing the transition at intermediate pressures. To date, the rate coefficient at the high-pressure limit has not been measured directly and the value of $k_1$ at 298 K of ~$2.4\times10^{-12}$ $cm^3$ molecule$^{-1}$ s$^{-1}$ at 96 bar (72000 Torr) of He, the highest pressure explored to date, is





still below the extrapolated $k_{1,\infty}$ value of $3.6\times10^{-12}$ cm$^3$ molecule$^{-1}$ s$^{-1}$ (Fulle et al., 1999). (Blitz et al., 2017a) derived values of

$k_{1,\infty}$ indirectly via measurements of the vibrational relaxation of OH in collision with SO$_2$, but obtained a much lower value of $k_{1,\infty} = 7.2\times10^{-13}$ cm$^3$ molecule$^{-1}$ s$^{-1}$. The value of $k_{1,\infty}$ presently recommended by IUPAC (last updated in November 2017) and NASA (last updated in May 2020) are $2.0\times10^{-12}$ and $1.7\times10^{-12}$ cm$^3$ molecule$^{-1}$ s$^{-1}$, respectively, falling between those of (Fulle et al., 1999) and (Blitz et al., 2017a).

From the studies reporting low-pressure limiting rate coefficients, largely obtained in flow-tube experiments, it is unclear

whether $k_{1,0}$ could be accurately derived through linear fitting to measured values of $k_1$ over a small pressure range, as measurements at the experimental pressures (a few Torr) were already impacted by fall-off. As discussed by Amedro et al. (2019), other factors such as wall-losses of OH add to the difficulty of deriving accurate low-pressure limiting rate coefficients in flow-tube experiments, especially at low temperatures.

Apart from N$_2$ and O$_2$, water vapour (H$_2$O) is a major atmospheric component. In particular, in the relatively warm boundary

layer, e.g. in the tropics, the mixing ratio of water vapour can exceed 5%. For termolecular reactions involving OH, H$_2$O may be expected to be a more efficient third-body quenching agent than N$_2$ and O$_2$ (Troe, 2003; Amedro et al., 2020) and the presence of water vapour can significantly enhance rate coefficients of third-body reactions in the low-pressure and fall-off regimes. A recent study (Amedro et al., 2020) demonstrated that neglecting to consider the influence of water vapour would lead to an underestimation in the rate coefficients of the OH+NO$_2$ reaction by ~ 10% in some parts of the lower atmosphere.

Given the similarities between the title reaction and the OH + NO$_2$ reaction, the participation of H$_2$O as a third-body quencher may also significantly enhance $k_1$, and such effects need to be examined experimentally.

The goals of this study were to address some of the shortcomings in the present dataset on the reaction of OH with SO$_2$ by accurately measuring $k_1$ in N$_2$ bath gas over a wide temperature/pressure range relevant for the atmosphere and to elucidate the role of water vapour as a third-body quencher on the title reaction. Such datasets are expected to yield a new

parameterization for $k_1$ with reduced uncertainties.

## 2 Experimental methods

Rate coefficients for the title reaction were derived using the pulsed-laser-photolysis (PLP)-laser induced fluorescence (LIF) technique under pseudo-first-order conditions ([SO$_2$] >> [OH], see Section 3.2). The concentrations of SO$_2$, as well as H$_2$O in the experiments exploring the effect of water vapour, were obtained online via optical absorption measurements.

### 2.1 PLP-LIF technique

Detailed descriptions of the PLP-LIF setup are given in previous publications (Wollenhaupt et al., 2000; Amedro et al., 2019). Briefly, the reaction took place in a jacketed quartz reactor with a volume of ~500 cm$^3$, the temperature inside the reactor was regulated by circulating ethanol (at 220 K) or 60:40 ethylene glycol mixture (250, 273 and 333 K) through an outer jacket. The temperature at the centre of the reaction volume was monitored by a J-type thermocouple before and after each experiment;



the pressure was measured using 100 and 1000 Torr capacitance manometers. To ensure that a fresh gas sample was photolysed at each laser pulse (laser frequency: 10 Hz) and to avoid accumulation of products, the average linear-velocity of gas flowing through the reactor was kept at $\sim 8-9$ cm s$^{-1}$ by adjusting the total volume flow rates according to the pressure and temperature in the reactor. For the vast majority of experiments, a KrF excimer laser (COMPex 205F, Coherent) provided pulses of 248 nm light ($\sim$20 ns) to photolyse $H_2O_2$ or $HNO_3$ for the generation of OH radicals in the vibrational ground-state (Schiffman et

al., 1993).

$H_2O_2 + h\nu$ (248 nm)          →          2 OH                                                                                  (R4)

$HNO_3 + h\nu$ (248 nm)          →          $OH + NO_2$                                                                        (R5)

In addition, a limited set of experiments were carried out in which HONO was photolysed at 351 nm (XeF excimer laser):

HONO + $h\nu$ (351 nm)          →          OH + NO                                                                            (R6)

OH radicals thus generated were excited at 282 nm ($A^2\Sigma$ ($v$=1) ← $X^2\Pi$ ($v$=0)) by a YAG-pumped dye-laser and the subsequent OH fluorescence was detected by a photomultiplier placed behind a 309 nm interference filter and a BG 26 glass cut-off filter. The photolysis laser fluence was measured by a Joule-meter placed behind the exit window of the reactor, and the shot-to-shot variation in the intensity of the dye-laser was monitored by a photodiode. The timing between the triggers of the photolysis and probe lasers was scanned using a digital delay generator; time-dependent OH profiles were obtained by accumulating the

fluorescence signals using a boxcar integrator. Fluorescence resulting from the excitation of $SO_2$ at 282 nm was also observed using this set-up, which results in a constant background signal during each experiment. Typically, 20 points were recorded before triggering the excimer laser to measure the background signal which also includes a component from electronic noise. The background signal was subtracted from the measured OH profile before further kinetic analyses.

**2.2 On-line optical absorption measurements**

The accurate determination of the rate coefficients under pseudo-first-order conditions requires reliable quantification of $SO_2$ concentrations ([$SO_2$]). In this work, online optical absorption cells were located both upstream and downstream of the reactor to monitor the $SO_2$ concentration at room temperature (298 K). Upstream of the reactor, light from a deuterium lamp was directed through a multi-pass absorption cell ($l$ = 110 cm $\times$8 = 880 cm) and detected by a low resolution ($\Delta\lambda$ = 2 nm) spectrograph (Ocean-Optics USB 2000). The measured optical density over the wavelength range of 240−325 nm was fit to a

reference spectrum (Manatt and Lane, 1993) to derive $SO_2$ concentrations. The second (downstream) optical absorption cell ($l$ = 34.8 cm) was equipped with a low-pressure Zinc lamp and monitored optical density at 213.86 nm to measure the concentration of $SO_2$ (see Section 3.1) and the OH precursors $H_2O_2$ and $HNO_3$. $SO_2$ concentrations were calculated using a 213.86 nm cross-section derived in separate experiments (see later). Concentrations of $H_2O_2$ and $HNO_3$ were calculated using $\sigma_{213.86\ nm}$ ($H_2O_2$) = $3.30 \times 10^{-19}$ (Vaghjiani and Ravishankara, 1989) and $\sigma_{213.86\ nm}$ ($HNO_3$) = $4.52 \times 10^{-19}$ (Dulitz et al., 2018).

Approximate, initial concentrations of OH radicals were calculated from the precursor concentrations and the excimer laser fluence as recorded by the Joule-meter. A third absorption cell ($l$ = 40 cm, located downstream of the 213.86 nm cell) equipped





with a low-pressure 184.95 nm Hg lamp as light source was additionally used in the experiments exploring the impact of water vapour. In this case, a cross section of $\sigma_{184.95\ nm}$ (H$_2$O) = 7.14 × 10$^{-20}$ (Cantrell et al., 1997) was used.

### 2.3. Chemicals

Nitrogen (N$_2$, 99.999 %) was supplied by Westfalen and used without further purification. SO$_2$ (Merck, 99.8%) was diluted in N$_2$ and stored in a stainless-steel cylinder. Hydrogen peroxide (H$_2$O$_2$, AppliChem, 35 %) was vacuum distilled to > 90 wt. % purity. Anhydrous HNO$_3$ was synthesized by mixing potassium nitrate (KNO$_3$, Sigma-Aldrich, 99%) and sulphuric acid (H$_2$SO$_4$, Roth, 98%), and condensing the HNO$_3$ vapour in a trap cooled with liquid nitrogen. Distilled water (Merck, liquid chromatography grade) was degassed before use.

### 3 Results and discussion

### 3.1. Quantification of SO$_2$ concentrations

SO$_2$ absorption spectra over 170−330 nm at room temperature are presented in **Fig. 1**, including a low-resolution ($\Delta\lambda$ = 0.1 nm) datasets reviewed and compiled by Manatt and Lane (1993) and a set of higher-resolution data ($\Delta\lambda$ = 2.5×10$^{-4}$ nm) spanning 198−325 nm reported by Stark and co-workers (Stark et al., 1999; Rufus et al., 2003). The two absorption bands

(240-330 nm and 170-230 nm) correspond to the $\tilde{B}(^1B_1) \leftarrow \tilde{X}(^1A_1)$ and the $\tilde{C}(^1B_2) \leftarrow \tilde{X}(^1A_1)$ transitions, respectively (Rollins et al., 2016). The high-resolution measurements by Stark et al. (1999) reveal very narrow rotational-vibrational lines in the 200−220 nm region, and cross-sections that are factors of 2-3 larger than those of Manatt and Lane (1993). Clearly, the use of an inappropriate cross section at the wavelength of our very narrow Zn-atomic emission line (~213.86 nm) could introduce large uncertainty in SO$_2$ concentration. For this reason, we derived an effective cross-section for our Zn-lamp emission by

comparing absorption measurements in the multi-pass cell and the 213.86 nm cell for a flowing gas-mixture of SO$_2$ in N$_2$. We refer to this cross-section as an "effective" cross-section as nearby, weaker atomic lines will also pass through the interference filter (214 ± 5 nm) used to isolate the 213.86 nm line. In **Fig. 1**, broadband (240−325 nm, resolution ~ 2 nm) absorption measurements recorded using the multi-pass cell are compared to the reference spectrum of Manatt and Lane (1993). SO$_2$ concentrations were determined through least squares fitting the measured optical density to the reference spectrum degraded

to the same resolution as our spectrometer. The effective cross section at 213.86 nm was then derived according to the Beer-Lambert law, taking into account pressure differences in the two absorption cells, as shown in **Fig. 2**. The slope of a linear regression through the data points is the effective cross-section of SO$_2$ at the emission wavelength of our low-pressure Zn-Lamp. The value obtained (4.00×10$^{-18}$ cm$^2$ molecule$^{-1}$) is very close (within ~ 3 %) to the value of 4.07×10$^{-18}$ cm$^2$ molecule$^{-1}$ listed at 213.86 nm by Stark et al. (1999) and the value of 3.87×10$^{-18}$ cm$^2$ molecule$^{-1}$ derived by Wine et al. (1984) using a

similar experimental setup. The high correlation coefficient (R$^2$ of 0.9984) for the linear regression of all data points obtained at different pressures (41 to 494 Torr) indicates that the effective value of $\sigma_{213.86}$(SO$_2$) is independent of pressure. SO$_2$



concentration measurements using the two absorption cells in the experiments at different temperatures are plotted in **Fig. 3**. The good agreement (slope very close to 1) between the measured (at room temperature) $SO_2$ concentrations upstream and downstream of the reactor (held at temperatures between 220 and 333 K) shows that no $SO_2$ is lost in transit (e.g. via adsorption

to surfaces or condensation) through the cold/hot reactor. The scatter in this plot is caused by small baseline shifts in the long path measurements upstream of the reactor.

### 3.2. Rate coefficients for the title reaction in $N_2$

Rate coefficients for the title reaction in $N_2$ bath were measured at pressures between 14 and 742 Torr at five different temperatures (220, 250, 273, 298 and 333 K). In deriving a parameterisation for the rate coefficient in air (see later), we assume

that, as is the case for the vast majority of termolecular reactions, $N_2$ and $O_2$ (the major components of air) have the same collisional quenching efficiency. In all experiments, the initial OH concentration was kept sufficiently low ($10^{11}-10^{12}$ molecule $cm^{-3}$) relative to that of $SO_2$ ($6\times10^{14}$ to $6\times10^{15}$ molecule $cm^{-3}$) so that pseudo-first-order conditions applied and the decay of OH may be described by:

$$[OH]_t = [OH]_0(-k't) \tag{1}$$

where $[OH]_t$ is the OH concentration at time $t$ after the photolysis laser pulse and $k'$ is the pseudo-first-order rate coefficient defined as

$$k' = k_1[SO_2] + k_d \tag{2}$$

$k_1$ (in $cm^3$ molecule$^{-1}$ s$^{-1}$) is the bimolecular rate coefficient for the title reaction and $k_d$ (in s$^{-1}$) accounts for the OH removal through reactions with $H_2O_2$ or $HNO_3$ (R7 or R8) as well as OH loss due to diffusion out of the reaction zone.

$OH + H_2O_2 \qquad \rightarrow \qquad H_2O + HO_2 \tag{R7}$

$OH + HNO_3 \qquad \rightarrow \qquad H_2O + NO_3 \tag{R8}$

**Fig. 4** displays a set of OH decay profiles at six different $SO_2$ concentrations ranging from 0 to $5.34\times10^{15}$ molecule $cm^{-3}$ at 298 K in 59.9 Torr $N_2$. $H_2O_2$ was used as OH precursor in this dataset and the initial OH concentration was ~ 4 $\times10^{11}$ molecule $cm^{-3}$. Each OH decay is the average of 20 measurements taken over a period of ~ 5 minutes. For each profile, the decay constant

$k'$ was obtained through least-squares fitting to Eq. (1). From each set of OH decays at a given temperature, pressure and bath-gas, the associated bimolecular rate coefficient $k_1$ was derived using Eq. (2) as shown in **Fig. 5** which plots $k'$ against $[SO_2]$ at 298 K at four different pressures.

Previous experimental studies have reported that the photo-excitation of $SO_2$ can have a large impact on the kinetics of OH loss. To circumvent this, optical filters containing $SO_2$ have been used to reduce the absorption of e.g. light from flash-lamps

(160-220 nm) (Paraskevopoulos et al., 1983; Wine et al., 1984). The 248 nm laser light used in the current set up is beyond the $SO_2$ photodissociation threshold of 219 nm so single-photon $SO_2$ photodissociation cannot affect the measurements for $k_1$. However, using a similar setup at 248 nm, (Blitz et al., 2017b, a) found evidence for two-photon dissociation of $SO_2$ in their experiments in He and reported a high-pressure limiting rate coefficient that is lower (by a factor of 2−5) than all other measurements or recommendations by evaluation panels. Blitz et al. (2017b) suggested that previous measurements were





biased by additional OH removal by radical-radical reactions initiated by the two-photon dissociation of $SO_2$. In order to

evaluate whether $SO_2$ photoexcitation could have impacted on our measurements of $k_1$ in $N_2$, we conducted measurements at

298 K and 14 Torr in $N_2$ with the excimer laser power varied by a factor of ~14. Over this energy range, the impact of any

two-photon processes would scale by a factor of ~ 200. $k_1$ was thus measured in a total of 10 experiments, for which $SO_2$

concentrations ranged from $1.2 \times 10^{15}$ to $6.0 \times 10^{15}$ molecule $cm^{-3}$, the concentration of $H_2O_2$ was kept at around $3 \times 10^{14}$ molecule

$cm^{-3}$, and the laser fluence was varied from ~0.5 to ~9.5 mJ $cm^{-2}$. Values of $k_1$ as a function of the 248 nm laser fluence are

displayed in **Fig. 6**. Our data clearly show that $k_1$ is independent of laser fluence (the average value is $1.29 \pm 0.05 \times 10^{-13}$ $cm^3$

molecule$^{-1}$ s$^{-1}$), suggesting that, under our experimental conditions in $N_2$ bath, secondary reactions between OH radicals

resulting from the photoexcitation of $SO_2$ are insignificant. This observation helps to rule out that single- or two-photon

processes involving $SO_2$ excitation or dissociation do not bias the OH decay and that reactions of OH with e.g. products of R1

(i.e. $HOSO_2$) or reaction with $HO_2$ and $NO_3$ (formed in R7 and R8) are unimportant. To examine the potential for two-photon

photolysis of $SO_2$ in He bath-gas, we added $H_2O$ to a reaction mixture of $H_2O_2$, $SO_2$ and He and observed non-exponential OH

kinetics, suggesting the intermediacy of $O(^1D)$:

$SO_2 + 2\ h\nu$        $\rightarrow$        $O(^1D) + SO$                                                    (R9)

$O(^1D) + H_2O$        $\rightarrow$        $2\ OH$                                                        (R10)

We emphasise that such effects were not seen in $N_2$ bath-gas in which $O(^1D)$ is rapidly quenched to less reactive $O(^3P)$.

To confirm beyond doubt that our measurements using 248 nm PLP are not biased by $SO_2$ excitation, an additional experiment

was performed (193.2 Torr and 298 K) using HONO photolysis at 351 nm  as OH precursor. At 351 nm the $SO_2$ absorption

cross-section is ~ 3 orders of magnitude lower than at 248 nm and $SO_2$ excitation is negligible. HONO was generated in-situ

by the dropwise addition of a 0.1 M $NaNO_2$ solution to a 20 wt. % $H_2SO_4$ solution and the characteristic bands at around 342,

354 and 368 nm (Stutz et al., 2000) were monitored by the multi-pass optical absorption cell. This setup provided sufficient

amounts of HONO (~$10^{14}$ molecule $cm^{-3}$) for kinetic measurements for about 1−1.5 hours after adding a few drops of the

$NaNO_2$ solution. Note that the concentration of $H_2O$ above the $H_2SO_4$ solution is very low, so that these can be considered to

be dry experiments (i.e. in $N_2$ bath-gas).

Unlike the $H_2O_2$ and $HNO_3$ sources of OH described above, the concentration of HONO was not stable over the time required

to measure a series of values of $k'$ in the presence of various amounts of $SO_2$. Therefore, measurements of $k'$ with and without

$SO_2$ were conducted intermittently. **Fig. 7** displays the measured first-order OH decay rate constants ($k'$) with different amounts

of $SO_2$ present in the system and over a period of ~ 2.5 hours. In the absence of $SO_2$ (blue symbols), the $k'$ decreased from an

initial value of ~ 2800 s$^{-1}$ to one of ~800 s$^{-1}$ at 5200 s. This is mainly due to the reaction of OH with HONO and impurities

such as $NO_2$ and NO. After ~5400 s a few drops of $NaNO_2$ were again added to the $H_2SO_4$ solution and the increase in [HONO]

was accompanied by an increase in $k'$. The decay in $k'$ over time was fit with a second-order polynomial function which, via

interpolation, was used to calculate the contribution of OH loss in the absence of $SO_2$ (i.e. $k_d$) from the individual values

obtained with $SO_2$ present. Based on the loss rate constants in the absence of $SO_2$ and the rate coefficient for reaction of OH



with HONO ($k_{11}$ (298 K) = $6.0 \times 10^{-12}$ cm$^3$molecule$^{-1}$s$^{-1}$) (Atkinson et al., 2004), we estimate the HONO concentrations to be ~$1-5 \times 10^{14}$ molecule cm$^{-3}$. .

HONO + OH        →        NO$_2$ + H$_2$O        (R11)

Combined with the photolysis laser fluence of around 1 mJ cm$^{-2}$, this results in an initial OH concentration of $0.3-1.5 \times 10^{11}$ molecule cm$^{-3}$. A total of five different SO$_2$ concentrations were used in these experiments, and at each SO$_2$ concentration the measurement of $k'$ was repeated four times. The averaged values of $k'$, after correction for the contribution from reactions of OH with HONO and other impurities as well as diffusion loss of OH, are plotted against SO$_2$ concentration in **Fig. 8**. A linear

regression yields a value of $k_1 = 4.91 \pm 0.13 \times 10^{-13}$ cm$^3$ molecule$^{-1}$ s$^{-1}$. A similar set of experiments at 273 K and 295 Torr yielded $k_1 = 8.44 \pm 0.19 \times 10^{-13}$ cm$^3$ molecule$^{-1}$ s$^{-1}$. Owing to the more convoluted experimental procedure and data analysis and also the larger OH losses in the absence of SO$_2$, (up to ~3000 s$^{-1}$) the overall uncertainty of the rate coefficient obtained in this manner is larger than those obtained using H$_2$O$_2$ and HNO$_3$ as OH precursors and the difference (~10 %) between the rate constant obtained using the 248 nm photouch lysis of H$_2$O$_2$ or HNO$_3$ or HONO as OH source is not considered significant. In

any case, this level of agreement (combined with the laser fluence variation described above) rules out a bias to $k_1$ of the magnitude reported by Blitz et al. (2017b) when working at 248 nm in N$_2$ bath-gas.

Our measurements of $k_1$ obtained in N$_2$ bath (in total > 100) are plotted against the N$_2$ concentration, at five different temperatures in **Fig. 9**. The rate coefficients and associated experimental conditions are listed in **Table S1** in the Supplementary Information**.** The rate coefficients obtained using H$_2$O$_2$ and HNO$_3$ as OH precursors (at 273 and 298 K) are indistinguishable

from each other, indicating that the rate coefficienty obtained are not influenced by secondary chemistry. The overall uncertainty in the values of $k_1$ (using H$_2$O$_2$ and HNO$_3$ precursors for OH) is estimated to be ~7%, which takes into account estimated systematic bias in the SO$_2$ cross-section at 213.86 nm.

The solid lines are fits to the experimental data according to the Troe formalism for termolecular reactions (Troe, 1983).

$$k_1(T,p) = \frac{k_{1,0}^{N2}\left(\frac{T}{300}\right)^{-n}[M]k_{1,\infty}\left(\frac{T}{300}\right)^{-m}}{k_{1,0}^{N2}\left(\frac{T}{300}\right)^{-n}[M]+k_{1,\infty}\left(\frac{T}{300}\right)^{-m}}F \qquad (3)$$

where $k_{1,0}^{N2}$ (cm$^6$ molecule$^{-2}$ s$^{-1}$) and $k_{1,\infty}$ (cm$^3$ molecule$^{-1}$ s$^{-1}$) are the high-pressure and low-pressure limiting rate coefficients, respectively; $T$ is the temperature (K), [M] is the molecular density (molecule cm$^{-3}$), $n$ and $m$ are temperature exponents. The broadening factor $F$ (which accounts for the lower rate coefficients measured in the fall-off regime than predictions by the Lindemann-Hinshelwood mechanism) is expressed as

$$\log F = \frac{\log F_C}{1+[\log(\frac{k_{1,0}^{N2}\left(\frac{T}{300}\right)^{-n}[M]}{k_{1,\infty}\left(\frac{T}{300}\right)^{-m}})/N]^2} \qquad (4)$$

where $N = 0.75$-$1.27\log F_C$ and $F_C$ is the broadening factor at the centre of the fall-off curve.

In order to reduce the number of variables when fitting, we assume that $k_{1,\infty}$ is independent of the temperature ($m = 0$). This assumption is reasonable as $m$ is expected to be small and the data at high pressures (neither in this work nor in the literature) do not accurately define this parameter. If all remaining variables ($k_{1,0}^{N2}$, $k_{1,\infty}$, $F_C$ and $m$) are allowed to float, the least-squares



optimization using Eq. (3) and Eq. (4) gives $k_{1,0}^{N2} = 3.03\times10^{-31}$ cm$^6$ molecule$^{-2}$ s$^{-1}$, $k_{1,\infty} = 2.00\times10^{-12}$ cm$^3$ molecule$^{-1}$ s$^{-1}$, $F_C = $

0.58 and $n = 4.10$ (see **Table 2,** Method 1). These parameters, with an R$^2$ correlation coefficient of over 0.994, accurately reproduce all our measurements in N$_2$ bath gas, as shown in **Fig. 9**.

As the range of temperatures encountered in the Earth's atmosphere is relative narrow, temperature-dependent forms of $F_C$ are no longer widely used in IUPAC evaluations, though a value of $F_C = \exp(-T/472)$ is still recommended for the reaction between OH and SO$_2$. We therefore also explored the effect of setting $F_C$ to $\exp(-T/472)$ and allowing a smaller set of variables, $k_{1,0}^{N2}$,

$k_{1,\infty}$ and $n$, to float while fitting. This results in a 20 % higher $k_{1,0}^{N2}$ of $3.60\times10^{-31}$ cm$^6$ molecule$^{-2}$ s$^{-1}$, an almost identical value of $k_{1,\infty} = 2.01\times10^{-12}$ cm$^3$ molecule$^{-1}$ s$^{-1}$ and $n = 2.86$ (**Table 2**, method 2). In addition, we examined the effect of varying the parameter $m$ (i.e. making $k_{1,\infty}$ temperature dependent) while $F_C$ was varied (but kept temperature independent). In this case we obtained $k_{1,\infty} = 2.03\times10^{-12}$ cm$^3$ molecule$^{-1}$ s$^{-1}$ with $m = -0.18$ and a lower value of $k_{1,0}^{N2}$ ($2.82\times10^{-31}$ cm$^6$ molecule$^{-2}$ s$^{-1}$) with $n = 4.34$ (**Table 2**, Method 3).

The quality of the fits obtained using Methods 1, 2 or 3 are very similar (see values for the residual standard deviation and correlation coefficients in **Table 2**), as highlighted in **Fig. S1** of the Supplementary Information where pressure and temperature dependent values of $k_1$ calculated using all three methods are plotted along with the experimental data. We also show in **Fig. S2** a plot of $k_1$ derived using each method versus altitude with the appropriate altitude dependent change in temperature and pressure for a standard atmosphere. Clearly, Methods 1 and 3, which have the lowest residual standard

deviations are in excellent agreement throughout the atmosphere, with slight differences to Method 2 in the stratosphere. Interestingly, the pressure and temperature dependence of $k_1$ cancel each other in the lowest 10 km of the Earth's atmosphere, so that $k_1$ is roughly constant at a value close to $1\times10^{-12}$ cm$^3$ molecule$^{-1}$ s$^{-1}$.

For the purpose of modelling the Earth's atmosphere, it is more important to ensure that the data at low and intermediate pressures and temperatures are correctly reproduced by the parameterization, with correct definition of $k_{1,\infty}$ of secondary

importance. For this reason, we have chosen to use the parameters derived in Method 1 to calculate $k_1$ and to compare with previous datasets and evaluations.

### 3.3. Comparison with previous parametrizations for N$_2$ bath-gas

Despite the importance of the title reaction in the atmosphere there were no prior temperature dependent measurements of $k_1$ in atmospherically relevant bath-gases. In experiments designed to define $k_{1,0}$, low-pressure flow-tube studies (Leu, 1982; Lee

et al., 1990) measured values of $k_1$ at N$_2$ pressures between 0.6 and 5 Torr. In order to access $k_{1,\infty}$, (Fulle et al., 1999) performed experiments in He bath-gas at pressures up to 96 bar. However, under all experimental conditions investigated so far, the title reaction is still in the fall-off regime and neither high- nor low- pressure limits for the title reactions have been attained directly, experimentally.

In **Table 2** we compare our values of $k_0$, $k_\infty$, $F_C$, $n$ and $m$ obtained in N$_2$ with the IUPAC and NASA expressions (both for N$_2$

bath-gas), as well as parameterizations reported by previous studies (Wine et al., 1984; Fulle et al., 1999; Blitz et al., 2017b),





which were mostly based on temperature-dependent measurements in other bath gases. The value of $k_{1,0}^{N2}$ derived in this work (Method 1) is close to those of $2.8\times10^{-31}$ and $2.9\times10^{-31}$ cm$^6$ molecule$^{-2}$ s$^{-1}$ preferred by IUPAC and NASA, respectively, and the current value of $k_\infty$ is identical to the recommendation of $2.0\times10^{-12}$ cm$^3$ molecule$^{-1}$ s$^{-1}$ by IUPAC; The current value of $n$ is equal to the vlue of 4.1 used in the NASA expression, but larger than the IUPAC preferred value of 2.6 unless we adopt the temperature dependent value for $F_C$ which IUPAC uses. In that case our value of $n = 2.86$ is close to that of IUPAC. This is readily understood as our value of $F_C = 0.58$ is very close to the "standard" NASA value of $F_C = 0.6$.

**Figure 10** provides a comparison between our data-points and parameterization (Method 1) with the literature data at 298 K obtained in N$_2$ (Leu, 1982; Lee et al., 1990; Paraskevopoulos et al., 1983; Wine et al., 1984) and with the IUPAC and NASA evaluations. In **Fig. S3** of the Supplementary Information we plot the ratios of literature rate constants obtained at 298 K in N$_2$ to our parameterisation. Also presented in **Fig. 10** is a modified parameterisation based on Blitz et al. (2017b) (master equation analyses of the (Paraskevopoulos et al., 1983) and (Wine et al., 1984) data sets). Note that the curve plotted cannot be reproduced using the parameters listed in **Table 2** of Blitz et al. (2017b) but is based on data (listed in Table 2) sent in a personal communication with Mark Blitz and takes care of various errors in the published analysis. The comparison in N$_2$ is restricted to 298 K as, prior to this study, all temperature dependent studies were performed in SiF$_6$, He or Ar.

In the common pressure range, (in the fall-off regime) our values of $k_1$ agree very well with those reported by Paraskevopoulos et al. (1983) and Wine et al. (1984). The fall-off curves described by our parameterization and the IUPAC and NASA recommendations are very similar over this pressure range, though our parameterization gives a slightly higher value of $k_1$ (298 K, 1 atm) ($1.04\times10^{-12}$ cm$^3$ molecule$^{-1}$ s$^{-1}$) than those proposed by IUPAC and NASA (9.38 and $9.50\times10^{-13}$ cm$^3$ molecule$^{-1}$ s$^{-1}$, respectively). In this context, we note that the value of $k_{1,\infty}$ chosen by IUPAC was a compromise between the rate coefficients measured at extended pressures of He by Fulle et al. (1999) and the data from Blitz et al. (2017a). These values are however very divergent with $k_{1,\infty} = 3.6\times10^{-12}$ cm$^3$ molecule$^{-1}$ s$^{-1}$ and $k_{1,\infty} = 7.5 \times 10^{-13}$ cm$^3$ molecule$^{-1}$ s$^{-1}$, respectively. We show below, that at 220 K our value of $k_1$ in 400 Torr of N$_2$ ($13.71 \pm 0.41 \times 10^{-13}$ cm$^3$ molecule$^{-1}$ s$^{-1}$) while clearly still in the fall-off regime, is a factor of two larger than the value of $k_{1,\infty} = 7.5 \times 10^{-13}$ cm$^3$ molecule$^{-1}$ s$^{-1}$ reported by Blitz et al. (2017a). Similarly, our value obtained using 351 nm photolysis of HONO as OH source (i.e. when SO$_2$ excitation can be ruled out) results in $k_1 = 8.44\pm0.19 \times 10^{-13}$ cm$^3$ molecule$^{-1}$ s$^{-1}$ in 295 Torr of N$_2$, again above the high-pressure limit reported by Blitz et al. (2017a). Blitz et al. (2017b) suggested that the larger literature values of $k_{1,\infty}$ available at the time of their study were a result of photo-excitation of SO$_2$. However, our results have clearly shown that a) such effects are absent in N$_2$ bath-gas when using 248 nm photolysis to generate OH and b) can be ruled out when working at 351 nm (see above). The excellent agreement between the present rate coefficients and those obtained previously in N$_2$ at 298 K (Paraskevopoulos et al., 1983; Wine et al., 1984) using completely different OH generation methods strongly suggest that data in N$_2$ are unaffected by such processes, though we cannot rule out that they are the cause of the large values of $k_{1,\infty}$ obtained by Fulle et al. (1999) in He. The present and previous datasets in N$_2$ indicate that $k_{1,\infty}$ is close to $2\times10^{-12}$ cm$^3$ molecule$^{-1}$ s$^{-1}$.



At lower pressures, the IUPAC, NASA and the present parameterizations capture the flow-tube measurements at 1 Torr and above (Leu, 1982; Lee et al., 1990), whereas at the very lowest pressures, there is substantial deviation. This is likely to reflect
bias in the flow-tube data caused e.g. by wall-losses of OH and large corrections for axial diffusion.

### 3.4. Influence of water vapour on $k_1$

To examine the effects of water vapour (e.g. as a 3rd-body quenching agent) on the kinetics of the title reaction, $k_1$ was measured in $N_2$-$H_2O$ mixtures at 273, 298 and 333 K. A total pressure of 50 Torr ($N_2 + H_2O$) was chosen for these experiments as sensitivity to a potential increase in $k_1$ through the presence of water is highest at conditions far from $k_{1,\infty}$. As described in
Section 2.2, both $SO_2$ (at 213.856 nm) and $H_2O$ (at 184.95 nm) were monitored online by optical absorption. The large absorption cross section of $HNO_3$ at 184.95 nm ($1.61\times10^{-17}$ cm$^2$ molecule$^{-1}$) (Dulitz et al., 2018) would hinder the accurate determination of the $H_2O$ concentration and $H_2O_2$ was therefore used as the OH precursor in all measurements in $N_2$-$H_2O$ bath-gas. The molar $H_2O$ mixing ratio in $N_2$ ($x_{H_2O}$) was varied up to 0.2 for measurements at 298 and 333 K, and up to 0.05 at 273 K to avoid condensation of water in the reactor and optical absorption cells. The results are summarized in **Table S2**, and
measured values of $k_1$ are plotted versus $x_{H_2O}$ in **Fig. 11** in which $k_1$ is seen to increase with the water content of the bath gas, indicating that $H_2O$ is a more efficient third-body quencher than $N_2$ for the title reaction. This is illustrated in **Fig. 12** where we plot the fall-off curves for $k_1$ in $N_2$ and $H_2O$ bath-gases. We note that, as for examining the water-vapour effect in the OH + $NO_2$ reaction (Amedro et al., 2020), continuous, in-situ measurement of the $SO_2$ concenration is critical to obtaining the correct result as $SO_2$ concentrations (i.e. optical density at 213.86 nm) decreased when $H_2O$ was added to the flowing $H_2O_2$ /
$SO_2$ / $N_2$ mixture. This presumably reflects losses of $SO_2$ on the glass surfaces of the apparatus on which $H_2O$ would have been adsorbed.

To evaluate the role of water on the rate coefficient of the title reaction, a parameterisation of the 3rd-body effect of $H_2O$ is required and we adopt the approach used in Amedro et al. (2020) for the OH + $NO_2$ reaction.

Extending Eq. (3), $k_1$ in $N_2$-$H_2O$ mixture can be expressed by

$$k(T,p) = \frac{(x_{N_2} k_{1,0}^{N2}\left(\frac{T}{300}\right)^{-n} + x_{H_2O} k_{1,0}^{H2O}\left(\frac{T}{300}\right)^{-o})[M] k_{1,\infty}\left(\frac{T}{300}\right)^{-m}}{x_{N_2} k_{1,0}^{N2}\left(\frac{T}{300}\right)^{-n} + x_{H_2O} k_{1,0}^{H2O}\left(\frac{T}{300}\right)^{-o})[M] + k_{1,\infty}\left(\frac{T}{300}\right)^{-m}} F \quad (5)$$

where $x_{H_2O}$ and $x_{N_2}$ are the mole fraction of $H_2O$ and $N_2$, $k_{1,0}^{H2O}$, is the low-pressure limiting rate coefficients (cm$^6$ molecule$^{-2}$ s$^{-1}$) in pure $H_2O$ and $o$ is a dimensionless temperature exponent. The broadening factor $F$ is then:

$$\log F = \frac{\log F_C}{1 + [\log(\frac{x_{N_2} k_{1,0}^{N2}\left(\frac{T}{300}\right)^{-n} + x_{H_2O} k_{1,0}^{H2O}\left(\frac{T}{300}\right)^{-o})[M]}{k_{1,\infty}\left(\frac{T}{300}\right)^{-m}})/N]^2} \quad (6)$$

Here, the low-pressure limiting rate coefficients in pure $N_2$ and pure $H_2O$ are linearly mixed and the same value of $F_C$ is
assumed for simplification. By inputting the values of $k_{1,0}^{N2}$, $k_{1,\infty}$, $F_C$ and $n$ listed in **Table 2** (Method 1), a multivariate, least-squares fit (solid lines in **Fig. 9**) results in $k_{1,0}^{H2O} = 1.65\times10^{-30}$ cm$^6$ molecule$^{-2}$ s$^{-1}$ and $o = 4.90$, indicating that $H_2O$, as a third-body collider, is at least five times more efficient than $N_2$.





We also consider the use of different values of $F_C$ for $N_2$ and $H_2O$, which may be more appropriate for bath-gases with distinctly different properties (Burke and Song, 2017) and adopted for the $OH+NO_2$ reaction in He-$H_2O$ mixtures (Amedro et al., 2020).

In the present case, however, $F_C$ for $N_2$ is already close to 0.6, so the differences are expected to be small and the use of the more complex expression for the purpose of atmospheric modelling of the reaction is not warranted. This is detailed in the Supplementary Information.

### 3.5. Atmospheric modelling of the OH + SO₂ reaction including the effect of water vapour

The chemistry and climate simulation model used is EMAC (ECHAM-MESSy) which uses the 5th generation European

Centre Hamburg general circulation model (ECHAM5, Roeckner et al. (2006)) as core atmospheric general circulation model (Jöckel et al., 2006; Jöckel et al., 2010). In this study we used EMAC (ECHAM5 version 5.3.02, MESSy version 2.55.0) at T63L47MA-resolution, i.e. with a spherical truncation of T63 (~1.8 by 1.8 degrees in latitude and longitude) with 47 vertical hybrid terrain following-pressure levels up to 0.01 hPa. The model was weakly nudged in spectral space, applying Newtonian relaxation of the parameters temperature, vorticity, divergence and surface pressure to meteorological reanalysis data (Jeuken

et al., 1996). The model set-up is identical to the simulation RED presented in Reifenberg et al. (2021) where the model was evaluated against an aircraft campaign over Europe. In addition, the model has been evaluated on many occasions (Pozzer et al., 2012a; Pozzer et al., 2012b; Yan et al., 2019). For additional references, see http://www.messy-interface.org. As in Reifenberg et al. (2021), EMAC was used in a chemical-transport model (Deckert et al., 2011) without feedbacks between photochemistry, radiation and atmospheric dynamics. In this work, we performed three identical simulations for the year 2019

but with three different parameterisations of $k_1$ from this work and from the IUPAC and NASA evaluations panels. For the simulations we assume that $O_2$ has the same collisional quenching efficiency as $N_2$ as is for case for nearly all termolecular processes of atmospheric importance.

In **Fig. 13**, we illustrate the impact of $H_2O$-vapour on the rate coefficient, by plotting the reduction in $k_1$ at the Earth's surface when setting $x_{H2O}$ to zero relative to using EMAC water-vapour fields. The greatest effect of water vapour on $k_1$ is found in

warm, tropical regions where an average underestimation of the rate coefficient by up to ≈ 5 % is found when $x_{H2O} = 0$. At higher/lower latitudes the effect is diminished and water vapour accounts for only a few percent of the overall rate coefficient at 40 ° N/S. The presence of water vapour does not impact significantly on values of $k_1$ above the boundary layer.

In **Fig. 14** we compare our new parameterisation with preferred parameterisations of the IUPAC and NASA evaluation panels and plot values of $k_1$(IUPAC) / $k_1$(This work) and $k_1$(NASA) / $k_1$(This work) at different altitudes and latitudes. We

parameterized $k_1$ using the expressions given in this work (Eqn. 5, **Table 2**) and in the latest evaluations of IUPAC (Iupac, 2021) and NASA (Burkholder et al., 2020). $k_1$(IUPAC) / $k_1$(This work) varies between 0.88 close to the surface to 0.72 at altitudes above ~ 30 km, whereas $k_1$(NASA) / $k_1$(This work) varies between 0.88 at the surface to 0.92 at ~ 30 km. Thus, while both evaluations under-predict $k_1$ by ≈ 12 % at the surface (partially related to the fact that they do not consider the effects of water-vapour), the NASA parameterisation does well in the lower stratosphere (under-predicting our result by less than 10%)





whereas the IUPAC parameters result in a rate coefficient that is too low by almost 30 %. At high altitudes the divergent rate coefficients recommended by the evaluation panels reflect the choice of experimental data used to derive the low-pressure limiting rate coefficient and its temperature dependence.

As the reaction between OH and $SO_2$ ultimately results in the formation of $H_2SO_4$ the atmospheric $H_2SO_4$ / $SO_2$ ratio is sensitive to the rate coefficient $k_1$, with an increase in $k_1$ resulting in a decrease in $SO_2$ and an increase in $H_2SO_4$, thus amplifying the

impact. In **Fig. 15** we plot zonally and yearly averaged model values of $\frac{H2SO4}{SO2}$ (IUPAC)$/\frac{H2SO4}{SO2}$ (this work) and $\frac{H2SO4}{SO2}$ (NASA)$/\frac{H2SO4}{SO2}$ (this work). Compared to the parameterization of $k_1$ in this work, the IUPAC evaluation returns $H_2SO_4$ / $SO_2$ ratios that are close to 0.9 at the surface but decrease to 0.7 in the lower to mid stratosphere at low latitudes. Again, the NASA parameterisation performs somewhat better, though here we also find an underestimation of the $H_2SO_4$ / $SO_2$ ratio of between 10 and 20 % throughout most of the atmosphere. The impact on the $H_2SO_4$ to $SO_2$ ratio is thus similar to

the change in the rate coefficients so that the expected amplification is not observed in the model. This is related to an increase in the sink term of $H_2SO_4$ (via nucleation) which counteracts the increase in its production rate.

The modelling studies indicate that use of IUPAC and NASA parameterizations result in very different values of $k_1$ in some parts of the atmosphere and will result in divergent predictions of partitioning of reactive sulphur gases between $SO_2$ and $H_2SO_4$. The present parameterisation, based on precise and accurate temperature dependent measurements in the fall-off regime

in $N_2$ does not rely on potentially erroneous data at very low and very high pressures and is expected to lead to more accurate values of $k_1$ for modelling the reaction in the Earth's atmosphere both at the surface (where the effect of water-vapour has been considered for the first time) and at the low pressures and temperatures prevalent in the UT-LS region.

## 5 Conclusions

Rate coefficients for the reaction of $SO_2$+OH ($k_1$) in fall-off regime were experimentally determined in a wide range of

pressures and temperatures relevant to the atmosphere. More than 100 individual measurements for $k_1$ were carried out in $N_2$ and $N_2$ / $H_2O$ bath gases using pulsed laser photolytic (PLP) generation of OH coupled to real-time detection of OH via laser-induced fluorescence (LIF). The presence of water-vapour was found to enhance the rate coefficient of the title reaction significantly, indicating that $H_2O$ is a more efficient 3rd- body collider than $N_2$ or $O_2$ (by a factor of > 5). Based on our comprehensive dataset in the fall-off regime, we derived a new parametrization of the rate coefficient which results in values

of $k_1$ that are larger than those preferred by the IUPAC and NASA panels, leading to a more rapid removal of $SO_2$ through gas-phase oxidation than previously assumed and thus to an underestimation of the $H_2SO_4$ / $SO_2$ ratio in nearly all regions of the Earth's atmosphere.

*Acknowledgements*. We thank Dr. Mark. Blitz for communicating revised parameters for the University of Leeds dataset on the title reaction.



*Data availability*. The rate coefficients measured during this experimental study are listed in the Supplementary information.

*Author contributions*. The experiments were carried out by WS with assistance from MB and JC. The data analysis was performed by WS. The global modelling was performed by AP. The manuscript was written by WS with assistance from JC and JL.

*Competing interests*. The authors declare that they have no conflict of interest.

*Financial support*. The article processing charges for this open access publication were covered by the Max Planck Society.

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





**Table 1. Temperature and pressure-dependent measurements of $k_1$**

| Reference | Technique | OH generation | [SO$_2$] measurement | Bath gas (Temp / K) | Pressure (Torr) |
|---|---|---|---|---|---|
| Leu (1982) | DF-RF | NO$_2$ + H | manometric | He (261-414) Ar (298) N$_2$ (298) O$_2$ (298) CO$_2$ (298) | < 10 |
| (Paraskevopoulos et al., 1983) | FP-RA | N$_2$O/H$_2$ + $h\nu$ | manometric | N$_2$ (297) | 50−760 |
| (Wine et al., 1984) | FP-RF | H$_2$O + $h\nu$ | optical absorption | He (300 K) Ar (260, 300, 360 and 420 K) N$_2$ (300 K) SiF$_6$ (260, 300, 360 and 420 K) | 13−696 |
| (Martin et al., 1986) | DF-EPR DF-MS | NO$_2$ + H | manometric | He (RT) | 1−6 |
| (Lee et al., 1990) | DF-RF | NO$_2$ + H | manometric | He (280-413) N$_2$ (298) O$_2$ (298) | < 6 |
| (Fulle et al., 1999) | FP-LIF | O$_3$/CH$_4$ + $h\nu$ | manometric | He (220-400) | 760−72000 |
| (Blitz et al., 2003) | LP-LIF | CHBr$_3$ H$_2$O$_2$ + $h\nu$ | manometric | He (295-673) | 100−200 |
| (Blitz et al., 2017a) | LP-LIF | SO$_2$ + H$_2$ + $h\nu$ (CH$_3$)$_3$COOH + h$\nu$ | manometric | He (295) | 25−303 |
| This work | LP-LIF | H$_2$O$_2$ + $h\nu$ HNO$_3$ + $h\nu$ HONO + $h\nu$ | optical absorption | N$_2$ (220-333) | 14-742 |

Notes: DF-RF = discharge flow- resonance fluorescence, FP-RA = flash photolysis- resonance absorption, FP-RF = flash photolysis-resonance fluorescence, DF-EPR-MS = discharge flow-electric paramagnetic resonance / mass spectrometry, FP-LIF = flash photolysis-laser induced fluorescence, LP-LIF = laser flash photolysis-laser-induced fluorescence. RT = room temperature.





**Table 2. Parameterisation of $k_1$ in N$_2$**

| | $k_{1,0}$ [1] | $n$ | $k_{1,\infty}$ [2] | $m$ | $F_C$ [3] | RSD [4] | CC [5] | Temp. range (K) |
|---|---|---|---|---|---|---|---|---|
| This work Method 1 | 3.03 (v) | 4.10 (v) | 2.00 (v) | 0 (f) | 0.58 (v) | 2.27 | 0.9943 | 220−333 |
| This work Method 2 | 3.60 (v) | 2.86 (v) | 2.01 (v) | 0 (f) | exp(-$T$/472) (f) | 2.44 | 0.9928 | 220−333 |
| This work Method 3 | 2.82 (v) | 4.34 (v) | 2.03 (v) | -0.18 (v) | 0.59 (v) | 2.27 | 0.9944 | 220−333 |
| Wine et al. (1984) | 5.76 | 2.57 | 1.26 | 0.7 | exp(-$T$/388) | | | 260−420 |
| Fulle et al. (1999) | 8.3exp(-360/T) | 3.3 | 12exp(-360/T) | 0 | 0.29+0.64exp(-$T$/300) | | | 220−400 |
| Blitz et al. (2017b)[6] | 10.6 | 3.53 | 0.79 | -0.10 | 0.386exp(-9.3×10$^{-5}$$T$) | | | 200−600 |
| IUPAC | 2.8 | 2.6 | 2.0 | 0 | exp(-$T$/472) | | | 200−400 |
| NASA[7] | 2.9 | 4.1 | 1.7 | -0.2 | 0.6 | | | -- |

Notes: [1] Units of $10^{-31}$ cm$^6$ molecule$^{-2}$ s$^{-1}$, [2] Units of $10^{-12}$ cm$^3$ molecule$^{-1}$ s$^{-1}$, [3] Temperature (T) in K. [4] RSD (residual standard deviation) for

the fitting is defined as $\sqrt{\sum(k_1 - k_{1_p})^2 / N - 2}$, $k_1$ and $k_{1_p}$ are the measured and the fitted rate coefficients and N is the total number of the

data points, with the unit of $10^{-14}$ cm$^3$ molecule$^{-1}$ s$^{-1}$. v = allowed to vary during fitting, f = fixed during fitting. [5]CC is the correlation

coefficient (R$^2$). [6] The parameters, which are different from those given in Blitz et al. (2017b) are from a personal communication with

Mark Blitz. The parameterization is based on the modified fall-off parameterisation in Troe and Ushakov (2014), details are given in the

Supplementary Information. [7] The simplified form of the Troe expression for termolecular reactions used by NASA can be found in the

Supplementary Information.





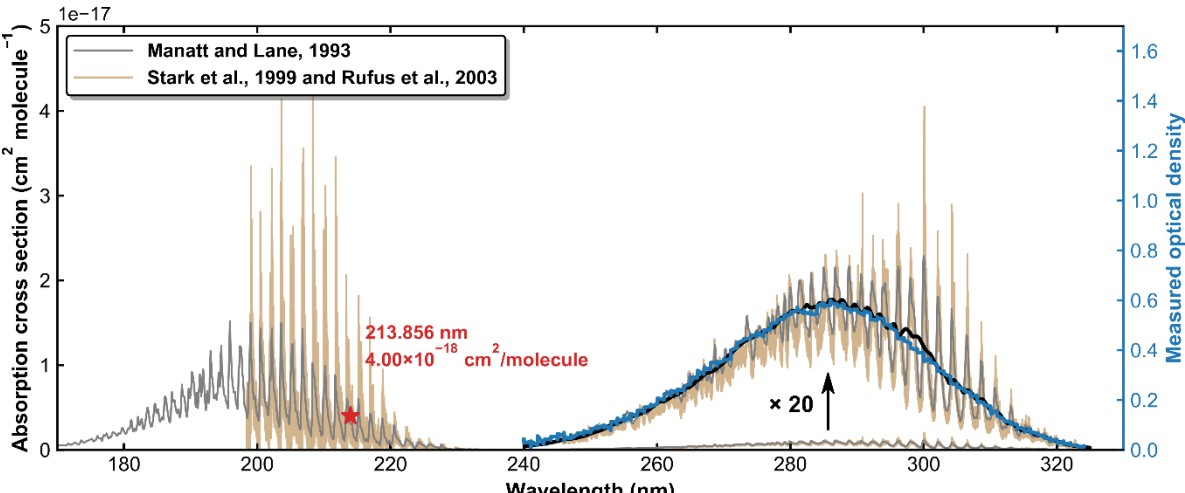

**Figure 1.** $SO_2$ absorption spectra reported by Manatt and Lane (1993) and Stark and co-workers (Stark et al., 1999; Rufus et al., 2003). The
black line is the result of degrading the resolution in the 240−325 nm region to that of our our optical density measurements (blue line).



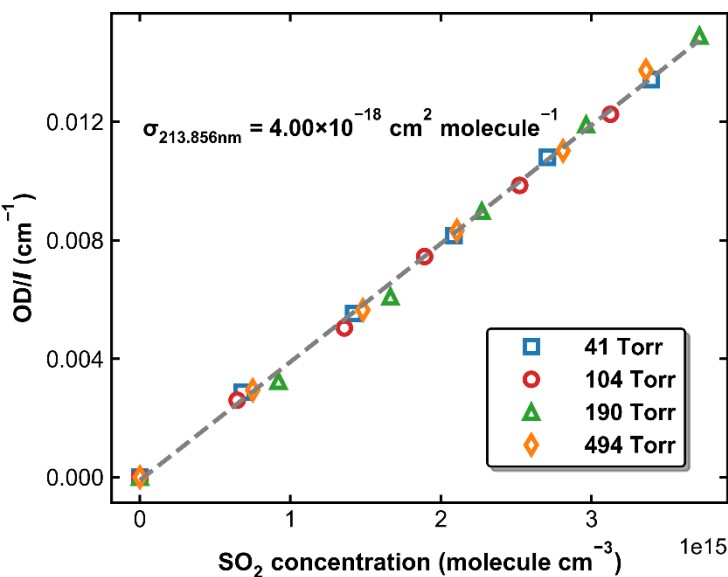

**Figure 2.** Normalized optical density in the 213.86 nm absorption cell as a function of SO$_2$ concentration measured using the multi-pass absorption cell. The experiments were performed at 298 K and pressures of 41, 104, 190 and 494 Torr. The grey dashed line is the linear regression, for which the slope is the effective cross section of SO$_2$ at 213.86 nm.





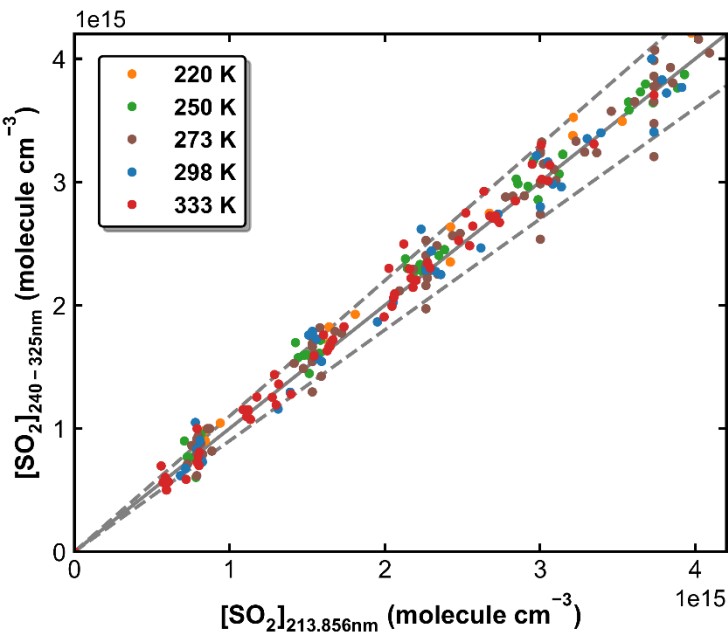

**Figure 3.** SO₂ concentrations measured in the experiments at different temperatures using the multi-pass absorption cell upstream of the reactor (*y* axis) and the 213.86 nm absorption cell downstream of the reactor (*x* axis). The solid line represents $y = x$, and the dashed lines are $y = 1.1x$ and $y = 0.9x$.





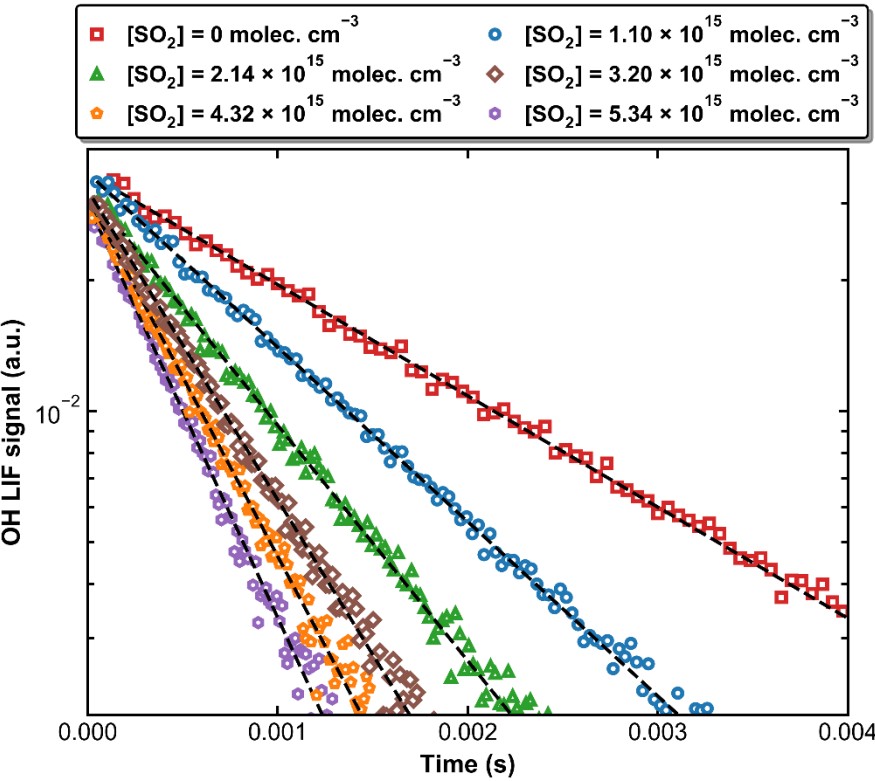

**Figure 4.** Exponential decay of OH at 298 K and 59.9 Torr (N$_2$ bath gas) in the presence of six different SO$_2$ concentrations. OH was generated by the photolysis of H$_2$O$_2$ at 248 nm The black dashed lines are fits using Eqn. (1).





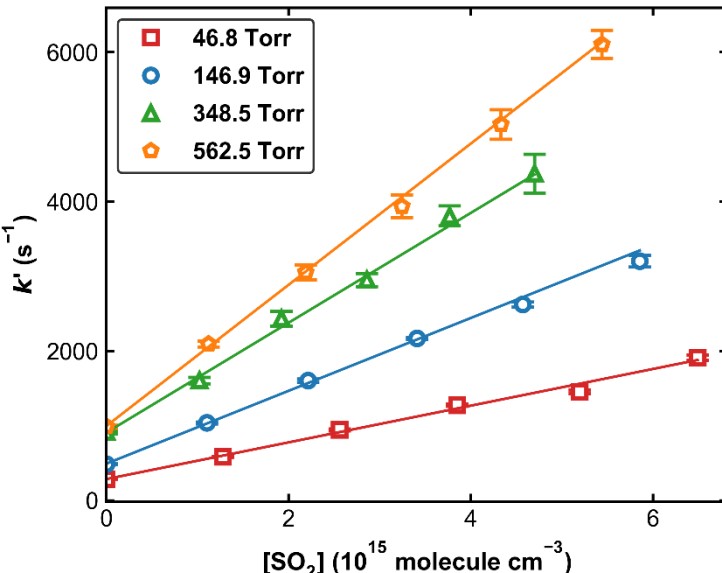

**Figure 5.** Pseudo-first-order rate coefficients ($k'$) as a function of SO$_2$ concentration ([SO$_2$]) at four different pressures at 298 K. Error bars

represent 2σ statistical uncertainties.The lines are weighted linear regressions.





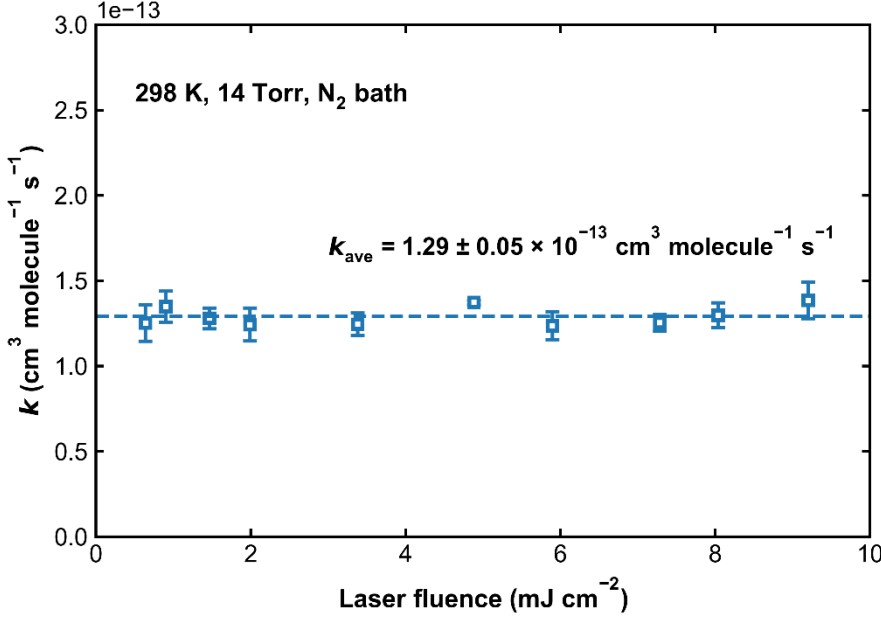

**Figure 6.** Values of $k_1$ measured at 298 K and in 14 Torr of $N_2$ bath gas. The 248 nm laser fluence was varied by a factor of ~ 14. The dashed line represents the average rate coefficient.






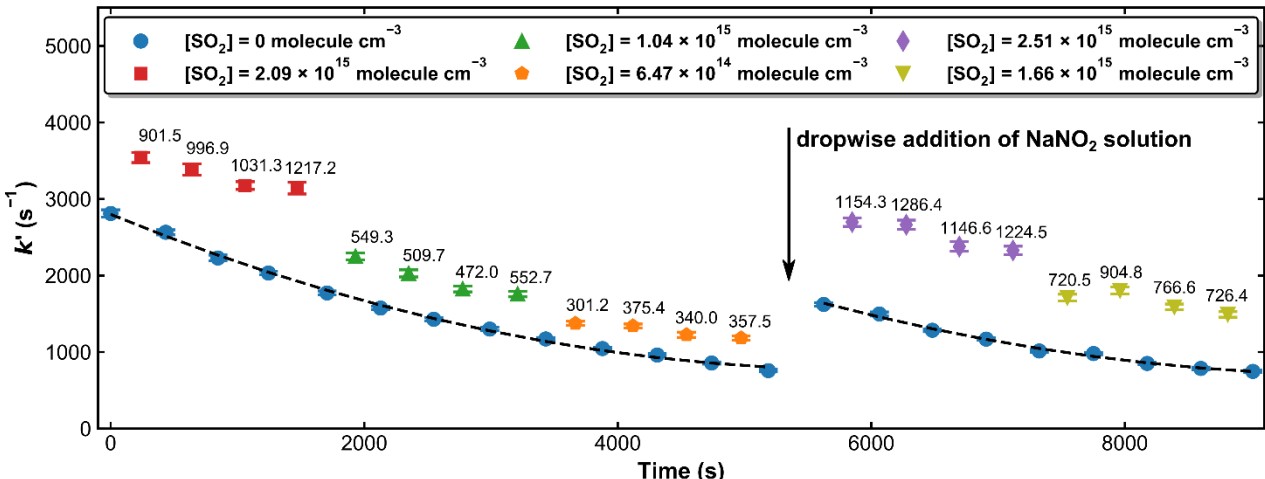

**Figure 7.** Pseudo-first-order rate coefficients ($k'$) at 193.2 Torr and 298 K whereby OH was generated by the 351 nm photolysis of HONO. The time dependent OH decay constants in the absence of $SO_2$ (blue symbols) were fit to a second-degree polynomial (black dashed lines). The number on the top of each data point when $SO_2$ is present (red, green, orange, purple and olive coloured symbols), represents the difference in $k'$ to that obtained (by interpolation) in the absence of $SO_2$. A few drops of the $NaNO_2$ solution was re-added after about 90 minutes to maintain OH levels and thus a good signal-to-noise ratio.



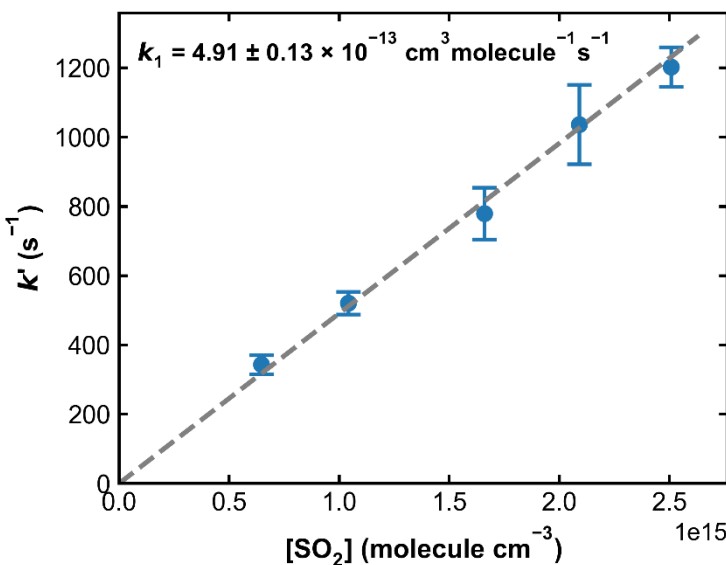

**Figure 8.** Pseudo-first-order rate coefficients ($k'$) as a function of the $SO_2$ concentration ($[SO_2]$) at 193.2 Torr $N_2$ and 298 K. OH was

generated through HONO photolysis at 351 nm. Each data point is an average over four individual measurements and the error bars represent $2\sigma$ statistical uncertainties. $k_1$ was obtained from the slope of the linear regression.



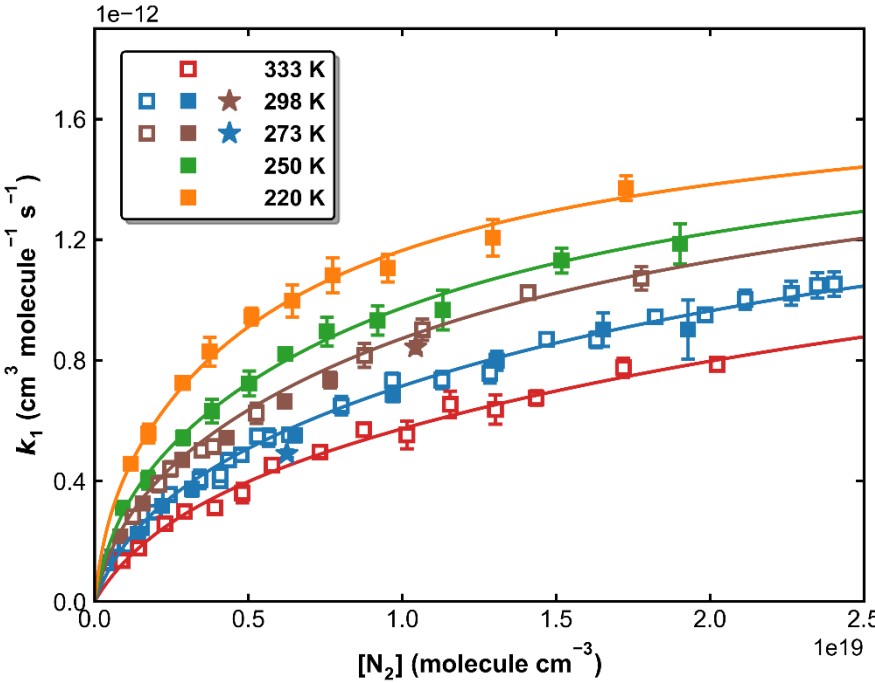

**Figure 9.** $k_1$ as a function of $N_2$ concentration in the fall-off regime at five different temperatures. Open squares, closed squares and stars represent data obtained using $H_2O_2$, $HNO_3$ and HONO as OH precursors, respectively. The error bars represent $2\sigma$ statistical uncertainties. The solid lines are the fits of the experimental data to Eqn. 3 with $k_0 = 3.03\times10^{-31}$ $cm^6$ $molecule^{-2}$ $s^{-1}$, $k_\infty = 2.00\times10^{-12}$ $cm^3$ $molecule^{-1}$ $s^{-1}$, $n = 4.10$, $m = 0$ and $F_C = 0.58$.


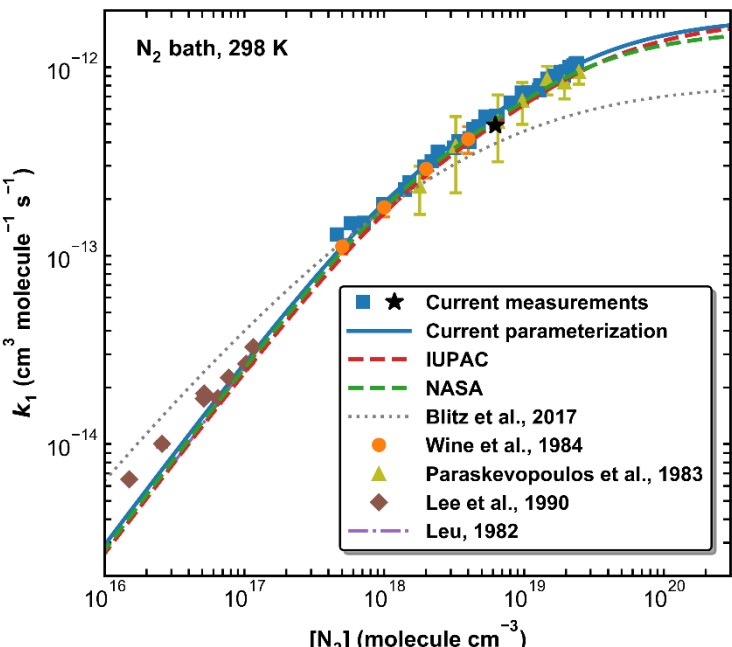

**Figure 10.** A comparison of previous measurements of $k_1$ at 298 K ($N_2$ bath-gas only) with our parameterisation and those of IUPAC, NASA and Blitz et al. (2017b) (see Table 2 for details). The black star represents the measurement using HONO photolysis at 351 nm to generate OH.





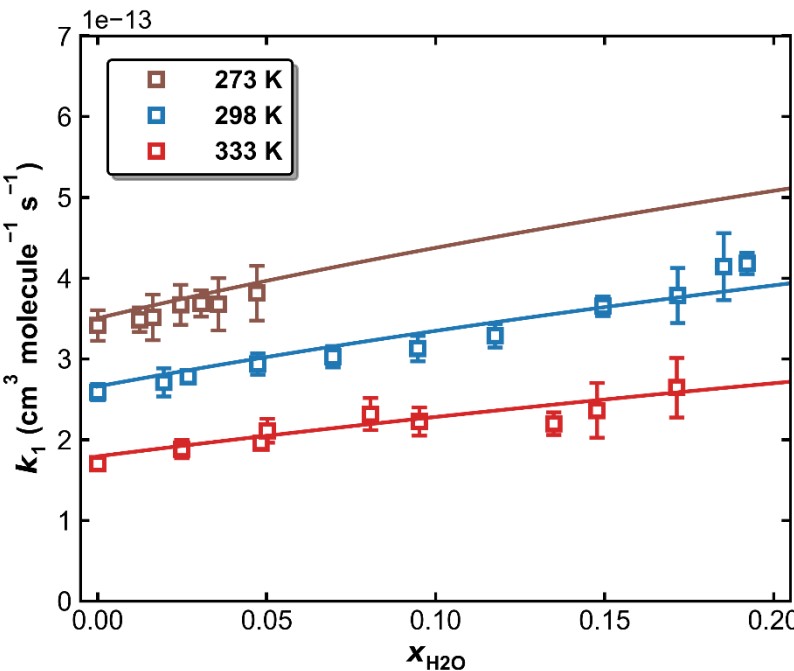

**Figure 11.** $k_1$ as a function of $x_{H_2O}$ in $N_2$-$H_2O$ bath gas at a total pressure of 50 Torr and different temperatures of 273, 298 and 333 K. The symbols represent measurements and the solid lines are fits to Eqn. (5) and (6) with $k_{1,\infty} = 2.00 \times 10^{-12}$ cm⁶ molecule⁻² s⁻¹, $k_{1,0}^{N_2} = 3.03 \times 10^{-31}$ cm⁶ molecule⁻² s⁻¹, $n = 4.10$, $m = 0$ and $F_C = 0.58$. The resulting parameters for water are $k_{1,0}^{H_2O} = 1.65 \times 10^{-30}$ cm⁶ molecule⁻² s⁻¹, $o = 4.90$.



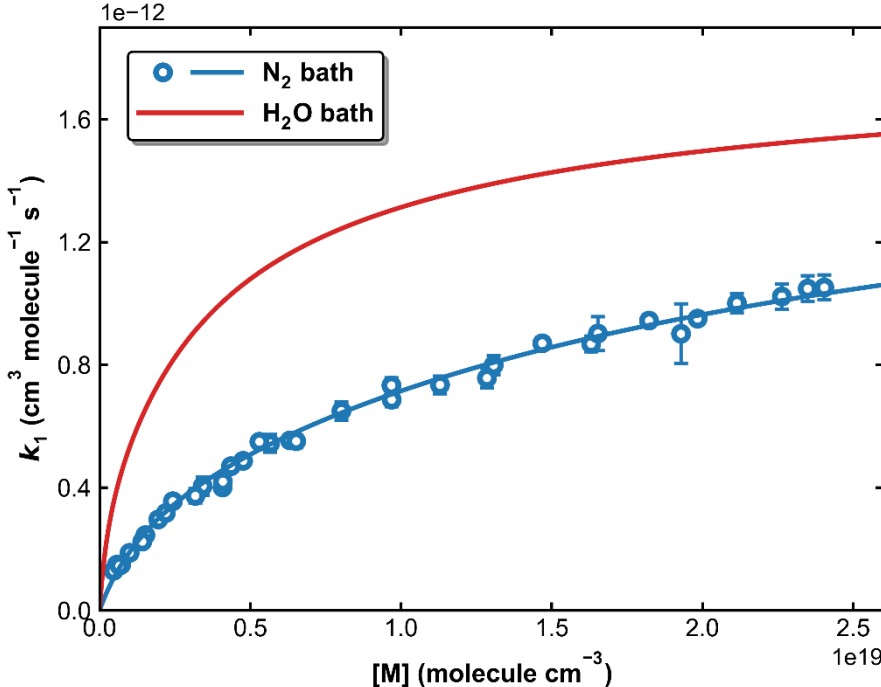

**Figure 12.** Fall-off curves for $k_1$ in $N_2$ and $H_2O$ bath-gases at 298 K. The solid lines are our preferred parameterization with $k_{1,\infty} = 2.0 \times 10^{-12}$ cm³ molecule⁻¹ s⁻¹ (independent of bath-gas), $k_{1,0}^{N2} = 3.03 \times 10^{-31}$ cm⁶ molecule⁻² s⁻¹ and $k_{1,0}^{H2O} = 1.65 \times 10^{-30}$ cm⁶ molecule⁻² s⁻¹, $F_C^{N2} = F_C^{H2O} = 0.58$.



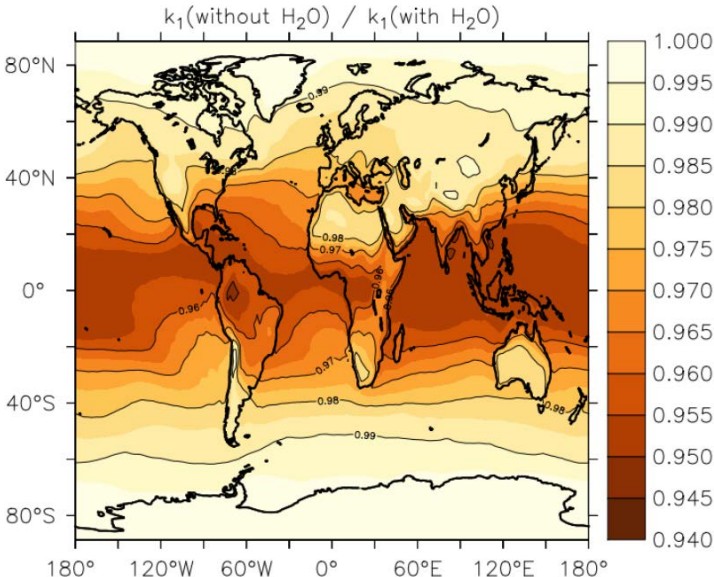

**Figure 13.** Annual average effect of $H_2O$ on $k_1$ expressed as the fractional change in the rate coefficient near the Earth's surface when setting the mole fraction of water vapour to zero in Eqn. 5 and 6.




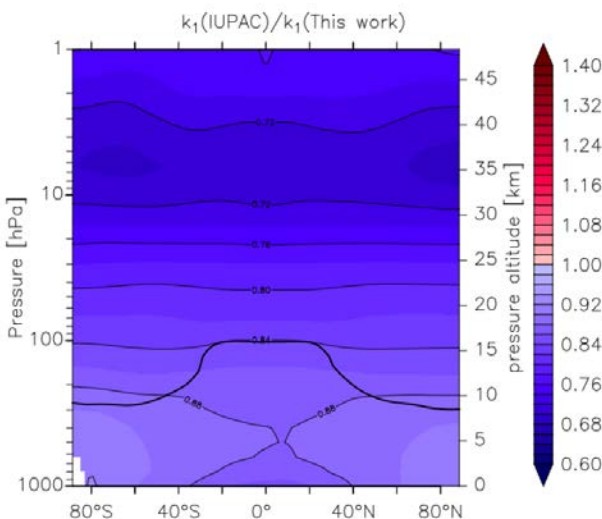

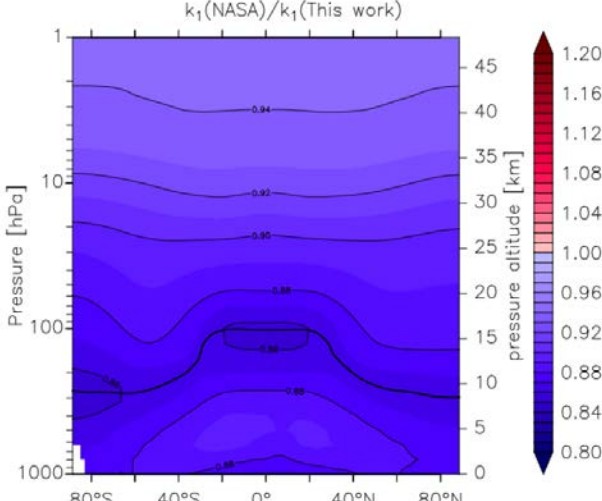

**Figure 14.** Global values of $\frac{k_1(\text{IUPAC})}{k_1(\text{This work})}$ (upper panel) and $\frac{k_1(\text{NASA})}{k_1(\text{This work})}$ (lower panel). $k_1$ was calculated using the parameters from this work and those presently recommended by the IUPAC and NASA data evaluation panels. The relatively thick black line between 300 and 100 hPa represents the mean model tropopause.




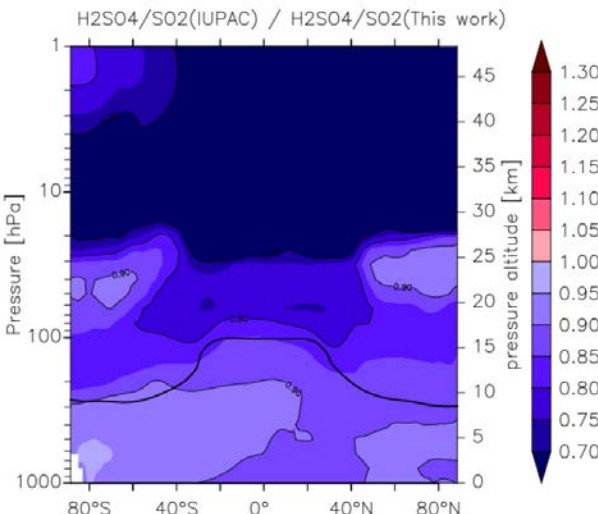

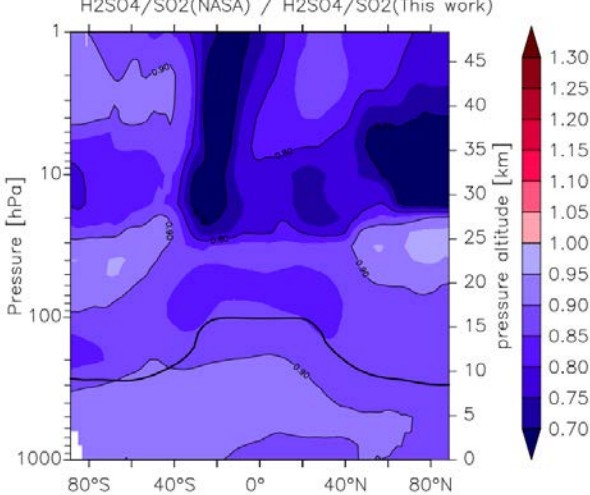

**Figure 15.** Effect of different parameterisations of $k_1$ on the global (zonal and yearly averaged) $H_2SO_4$ to $SO_2$ ratio. The upper

panel plots $\frac{H2SO4}{SO2}$ (IUPAC)/$\frac{H2SO4}{SO2}$ (this work), the lower panel plots $\frac{H2SO4}{SO2}$ (NASA)/$\frac{H2SO4}{SO2}$ (this work). The relatively thick

black line between 300 and 100 hPa represents the mean model tropopause.