# Peer review of "Kinetics of $OH + SO_2 + M$ : Temperature-dependent rate coefficients in the fall-off regime and the influence of water vapour"

_Atmospheric Chemistry and Physics, 2021_

## Author Response (AR1)

**Referee 1**

In the following, we list the comment (black), our reply (blue) and indicate which changes have been made to the manuscript (red).

The manuscript reports on an experimental kinetic study of the $OH + SO_2 + M$ reaction. The experiments were performed in a pulsed laser photolysis (PLP)/laser-induced fluorescence (LIF) setup with OH production from photolysis of $H_2O_2$, $HNO_3$, or HONO and time-resolved detection of OH with LIF. Rate constants were determined under pseudo-first order conditions with respect to OH. The excess concentration of $SO_2$, which is critical for the second- and third-order rate constants, was carefully determined with UV absorption spectroscopy. Neat $N_2$ or mixtures of $N_2$ with $H_2O$ were used as bath gases, and it was found that $H_2O$ is a particular efficient collider leading to a notably increased rate constant. In general the rate constants were found to be in the falloff range at the chosen conditions ($T = 220$–$333$ K, $p = 14$–$742$ Torr), and their pressure dependence was parameterized in terms of Troe expressions. These results were implemented in the chemistry part of an atmospheric general circulation model to assess the influence of atmospheric water content. It was found that the atmospheric lifetime of $SO_2$ is probably lower than previously assumed in nearly all regions of the atmosphere. Overall, this manuscript is a fine piece of work combining very carefully performed laboratory experiments with adequate parameterizations of rate constants and atmospheric modeling calculations. The scientific problem addressed is timely, and the methods used are adequate and state-of-the-art. The results are carefully discussed and compared with those from other works, and the paper is excellently written. There is almost nothing to complain. Hence, I recommend acceptance of the manuscript for ACP after minor, mainly technical revisions.
We thank the referee for this highly positive assessment of our manuscript.

General: The authors should carefully check the consistent use of rate constant symbols. Sometimes the temperature dependence $(T/300)^n$ is included in the rate constant, sometimes it is not (cf. e.g. the use of $k_{1,0}^{N2}$ in the abstract and introduction section and its use in eqs. (3–6). We now use rate constant symbols consistently throughout the manuscript, using, where possible the full form (see example below)
....with $k_{1,0}^{H2O} = 1.65\times10^{-30}\ (T/300\ \mathrm{K})^{-4.90}\ \mathrm{cm^6\ molecule^{-2}\ s^{-1}}$

Also, in the abstract, nothing is said about the $T$ dependence of $k_{1,0}^{H2O}$ whereas the temperature exponent $o = 4.90$ on page 11 (bottom). This must be corrected. Also better use $(T/300\ \mathrm{K})^n$ instead of $(T/300)^n$ etc.
We now list the temperature dependence in the abstract and have added "K" to the $(T/300\ \mathrm{K})$ term
....with $k_{1,0}^{H2O} = 1.65\times10^{-30}\ (T/300\ \mathrm{K})^{-4.90}\ \mathrm{cm^6\ molecule^{-2}\ s^{-1}}$

lines 58, 183, 312: Shouldn't "photo-excitation of $SO_2$" better read "photo-dissociation of $SO_2$"?
No, photoexcitation (which may or may-nor result in dissociation) was deliberately chosen as the do not want to rule out the possibility that non-dissociated, excited states of $SO_2$ also react.

line 61: Check parentheses in rate constant symbols.
Typos corrected
Line 61 (now 62): high-pressure ($k_{1,\infty}$) and low-pressure ($k_{1,0}$)

line 97: In line 92, the volume of the quartz reactor is given (500 cm$^3$). So it would be better to give the flow rate for typical $T$ and $p$ also in units cm$^3$ s$^{-1}$ instead of cm s$^{-1}$. The reader does not know the length of the reactor. Are with these flow rates really fresh(!) gas samples photolyzed at each laser pulse (with 10 Hz repetition rate).

We did not mention that the axis of flow-direction and laser-beams are prependicular to each other and that the reactor is tubular. We now clarify this:

The average linear-velocity of gas flowing through the tubular reactor was kept at ~ 8−9 cm s$^{-1}$ by adjusting the total volume flow rates. As the flow direction and laser-beams (0.8 cm diameter) are perpendicular to each other, a linear velocity of over 8 cm$^{-1}$ ensures that a fresh gas sample was photolyzed at each laser pulse (10 Hz) and the volume of gas imaged onto the PMT is replenished between pulses.

eq. (5): One ( in the denominator is missing.

The typo has been corrected.

Line 345: $k(T, p) = \dfrac{\left(x_{N_2} k_{1,0}^{N2}\left(\frac{T}{300}\right)^{-n} + x_{H_2O} k_{1,0}^{H2O}\left(\frac{T}{300}\right)^{-o}\right)[M] k_{1,\infty}\left(\frac{T}{300}\right)^{-m}}{\left(x_{N_2} k_{1,0}^{N2}\left(\frac{T}{300}\right)^{-n} + x_{H_2O} k_{1,0}^{H2O}\left(\frac{T}{300}\right)^{-o}\right)[M] + k_{1,\infty}\left(\frac{T}{300}\right)^{-m}} F$

**Referee 2**

In the following, we list the comment (black), our reply (blue) and indicate which changes have been made to the manuscript (red).

This manuscript presents experimentally determined rate coefficients for OH+SO2 over a range of atmospherically relevant conditions (T = 220–333 K, p = 14–742 Torr $N_2$ and $xH_2O$ = 0 – 0.2). This reaction plays a key role in the sulfur-controlled particle formation in the Earth's atmosphere, and, similar to research from the same group into OH + $NO_2$, warranted updating to include the rate coefficient enhancement when considering $H_2O$ as a third body collision partner of the $HOSO_2^*$ association complex. There is a large body of experimental work presented herein using a combination of $N_2$ and $H_2O$ bath gases to parameterize the rate coefficient. The authors find that $H_2O$ is >5 times more efficient a collision partner compared to $N_2$, which is significant. Much care was taken to characterize the negligible effect of $SO_2$ photolysis at 248 nm on the rate coefficient measurement which possibly affected previous determinations of the rate coefficient in He. This included laser energy dependencies, spectroscopic $[SO_2]$ measurements and utilizing HONO as an alternative OH source at 351 nm; a very thorough and meticulous investigation.
This work is rounded off by looking at the overall impact this new parameterization has on the determination of the title rate coefficient, where the authors show there is a significant discrepancy compared to current parameterizations in the recommended IUPAC/NASA literature throughout the troposphere and stratosphere.
I recommend this article for publication with only a few (very) minor comments below.
We appreciate the encouraging positive comments on our manuscript from this referee.

Table 1/Line 255: I think the inclusion of RSD is a good tool for us to judge the goodness-of-fit and should be potentially adopted by others, however $R^2$ is somewhat meaningless for non-linear regressions. Additionally, the correlation coefficient, R, and the coefficient of determination, $R^2$, are not the same, as stated in the caption.
We have excluded the $R^2$ column from **Table 2** and use RSD instead of $R^2$ in the main text when mentioning the quality of non-linear regression.
Line 159: "…The high coefficient of determination ($R^2$ of 0.9984)…"
Lines 259–260: "…with a small residual standard deviation (RSD) of $2.27\times10^{-14}$ cm$^3$ molecule$^{-1}$ s$^{-1}$ …"

Figure 9/S1: Perhaps the red open-square in the legend could be made to line up with the other open squares for clarity?
We have followed the referee's suggested and updated the legends.

Figure S2: These altitude profiles represent the change in the "dry" rate coefficient, correct? I think it would be good to clarify the ~5% increase in the rate coefficient before the effect of water vapor is included when comparing the data to the current IUPAC/NASA profiles. This is really only important in the lower part of the atmosphere, but would highlight the compounding effect of the water vapor collision efficiency in the following modelling section.
Yes, the profiles represent rate coefficients in the absence of $H_2O$. We have clarified this in the caption of **Fig. S2** "Parameterized $k_1$ in $N_2$ bath gas". The relevant discussion has been modified.
Lines 277–278: "so that $k_1$ is roughly constant at a value close to $1\times10^{-12}$ cm$^3$ molecule$^-$

$^1$ s$^{-1}$, which is about 5% higher than the IUPAC and NASA recommendations."

Lines 385–386: "Thus, while both evaluations under-predict $k_1$ by $\approx 12$ % at the Earth's surface (a combined consequence of their lower values of $k_1$ in $N_2$ (dry air) and neglecting water vapour effects), the NASA parameterization does well in the lower stratosphere (under-predicting our result by less than 10%) whereas the IUPAC parameters result in a rate coefficient that is too low by almost 30 %"

L368 and Fig 13. Over what altitude range are these results integrated when considered "at the Earth's surface" and/or "near the Earth's surface", as quoted from the main text and figure caption respectively?

When referring to the Earth's surface we mean the lowest 1-2 km of the atmsophere, which we now define at first usage.

"...by plotting the reduction in $k_1$ at the Earth's surface (lowest 1-2 km of the atmosphere) when setting.."